# DNA metabarcoding reveals the dietary profiles of a benthic marine crustacean, *Nephrops norvegicus*

**Peter Shum**[1,2]*, **Janine Wäge-Recchioni**[2,3], **Graham S. Sellers**[2], **Magnus L. Johnson**[4], **Domino A. Joyce**[2]

**1** Faculty of Science, Liverpool John Moores University, Liverpool, United Kingdom, **2** School of Natural Sciences, University of Hull, Hull, United Kingdom, **3** Leibniz Institute for Baltic Sea Research Warnemünde (IOW), Rostock, Germany, **4** School of Environmental Sciences, University of Hull, Hull, United Kingdom

* P.Shum@ljmu.ac.uk

**Data Availability Statement:** Raw Illumina sequences can be found on NCBI's SRA database BioProject ID: PRJNA911567. The data set including MOTUs, taxonomic assignment,

## Abstract

Norwegian lobster, *Nephrops norvegicus*, are a generalist scavenger and predator capable of short foraging excursions but can also suspension feed. Existing knowledge about their diet relies on a combination of methods including morphology-based stomach content analysis and stable isotopes, which often lack the resolution to distinguish prey items to species level particularly in species that thoroughly masticate their prey. DNA metabarcoding overcomes many of the challenges associated with traditional methods and it is an attractive approach to study the dietary profiles of animals. Here, we present the diet of the commercially valuable *Nephrops norvegicus* using DNA metabarcoding of gut contents. Despite difficulties associated with host amplification, our cytochrome oxidase I (COI) molecular assay successfully achieves higher resolution information than traditional approaches. We detected taxa that were likely consumed during different feeding strategies. Dinoflagellata, Chlorophyta and Bacillariophyta accounted for almost 50% of the prey items consumed, and are associated with suspension feeding, while fish with high fisheries discard rates were detected which are linked to active foraging. In addition, we were able to characterise biodiversity patterns by considering *Nephrops* as natural samplers, as well as detecting parasitic dinoflagellates (e.g., *Hematodinium* sp.), which are known to influence burrow related behaviour in infected individuals in over 50% of the samples. The metabarcoding data presented here greatly enhances a better understanding of a species' ecological role and could be applied as a routine procedure in future studies for proper consideration in the management and decision-making of fisheries.

## Introduction

Marine animals possess a diverse repertoire of feeding strategies linked to individual and species-specific foraging behaviour in complex, multifood environments. They engage in food acquisition at individual, population and community levels that shape ecosystem functioning across trophic levels. This influences biotic interactions and behaviour among species which

abundances as well as OBITools bioinformatics scripts, R scripts, and sample barcodes are available in the GitHub repository (https://github.com/shump2/Nephrops-diet).

**Funding:** The author(s) received no specific funding for this work.

**Competing interests:** The authors have declared that no competing interests exist.

drives phenotypic selection and eco-evolutionary feedbacks [1–3]. At an individual level, feeding is linked to nutrition and ecophysiology as the quantity and quality of food resources regulate individual survival, growth and fecundity. The ability to monitor these dietary profiles of animals, particularly those of commercial importance, across spatial scales is fundamental to understanding how environmental and anthropogenic activities influence nutrition. An accurate and comprehensive dietary matrix of species in marine ecosystems is an essential foundation for ecological fisheries models that are necessary for developing a better understanding of likely impacts of climate change on marine systems [4].

The methodology for studying the diet of animals is varied and considers the type of sample collection, sample processing, and the identification and quantification of prey consumption. In the marine environment, species are rarely observed to forage directly and most studies depend on the identification of prey remains in stomach contents or faeces to determine the prey items being consumed [5]. However, obtaining detailed diet information is challenging for many species because of the effort required to directly observe and physically identify food items from stomach contents [5]. This is particularly problematic in understudied species as traditional stomach content analyses rely on extensive experience to identify species specific characteristics of hard parts (e.g., otoliths, scales, cleithra, carapace) and soft contents with the aid of good references to identify items from digested, broken and finely comminuted material [6]. But diet items that have been recently consumed can be rapidly digested and become quickly indiscernible which can underestimate dietary composition. This lack of consistent information can negatively impact food web models in estimating annual catch and consumption by predators [7] and could lead to unexpected and undesirable management outcomes [8]. DNA barcoding using the universal cytochrome oxidase I (COI, ~650bp) gene can improve detection of otherwise unidentifiable individual prey items, however there is reduced success for highly digested material with time and resources needed to process separate items [9]. Another popular application is the use of carbon and nitrogen isotopic signatures (i.e., stable isotopes) in tissues to measure the differential assimilation of dietary components which can overcome some of the shortcomings of traditional dietary studies [10]. For example, Wieczorek *et al.* [11] studied the diet of the lesser spotted dogfish using the stomach content analysis and stable isotopes. They found that stomach content analysis presents a snapshot of the diet that overestimate hard-bodied prey species while stable isotopes revealed that soft-bodied filter feeders were by far the most important diet items, accounting for approximately 76% of the energy assimilated. This highlights the advantage of complementary approaches to provide a better overview of a species trophic position.

While stable isotope analysis is a powerful method to complement stomach content data in estimating dietary profiles, it lacks the resolution to accurately recover species level information [12, 13]. An alternative method that has quickly gained momentum in trophic ecology studies is DNA metabarcoding. DNA metabarcoding embraces DNA barcoding of short DNA fragments and next generation sequencing to identify species information from a variety of sample types, and facilitates the detection of small, soft-bodied or cryptic species which might be overlooked during traditional diet analysis [14]. The power and utility of DNA metabarcoding has become an attractive tool to identify food DNA consumed by animals to reveal dietary habits [15], parasite load [16], trophic niches [17], and local natural biodiversity [18].

*Nephrops norvegicus*, (also referred to as the Norwegian lobster, Dublin Bay Prawn or scampi, *Nephrops* hereafter), is a benthic decapod crustacean and is one of the most important economically valued fisheries in Europe generating a value of 50 M€, making it the second most valuable landed species in the North Sea (NS) and Eastern Arctic region in 2019 [19]. They are distributed over semi-isolated mud patches throughout the North-Eastern Atlantic Ocean in the North Sea as far south as the Canary Islands and extending into the Eastern

Mediterranean Sea [20]. *Nephrops* are thought to be crepuscular opportunistic predators and scavengers, found to feed on fish, crustaceans, molluscs and other taxa [21]. *Nephrops* also possess a complex mode of feeding by capturing and ingesting suspended particulate organic matter (i.e., $POM_{susp}$) from the water column, i.e., suspension feeding. Suspension feeding is thought to be used especially by females for surviving starvation during the long breeding period which lasts from late spring to early autumn when they are restricted to their burrows [22]. Santana et al. [23] revealed the importance of suspension feeding through stable isotopes and found that half of their diet was made up of suspended particulate organic matter alone. This study collected samples during the spring, coinciding with the breeding season, but found no differences between male and female *Nephrops* in terms of their feeding habits. Fish were shown to be another important food source, but it is unclear which species contribute to their diet or whether their diet is supplemented by discards arising from inshore fisheries [23]. The link between diet and behaviour in *Nephrops* is particularly important to fisheries as when they are in their burrows, they cannot be captured by fishers. The fishing fleet is aware that *Nephrops*, as well as having a crepuscular emergence habit, will appear and disappear en masse in different grounds at different times of year, and fishers will move among the discrete fishing grounds (functional units) as they become available [24].

Here we applied DNA metabarcoding to characterise the gut contents of *Nephrops norvegicus* from specimens collected in the North and Irish Seas. *Nephrops* are a generalist forager and highly commercial benthic crustacean. Utilizing a molecular approach, we can consider them as unique natural biodiversity samplers, offering valuable insights into their ecosystem. Therefore, we aim to i) characterise their feeding strategy using DNA metabarcoding of digested material in the gut and ii) examine the biodiversity of prey consumed.

## Materials and methods

### Sample collection and processing

A total of 207 *Nephrops norvegicus* specimens were collected on board commercial fishing vessels in the East (n = 77) and West (n = 68) of the North Sea (NS-East and NS-West respectively)—Fladen Grounds and the Irish Sea (n = 63) in January 2016. Given the particularities of these catches, the specific locations were not provided to us. Instead, we received only broad, generalised areas where the collections were made, reflecting the common practice in commercial *Nephrops* fishing. Specimens were collected from the seafloor and were stored on ice at sea and stored at -80˚C in the lab prior to dissection. In the lab, each specimen was thawed on ice after which the total contents of the gastro-intestinal tract was placed into a DNeasy PowerSoil tube using sterile forceps and DNA extraction was performed following the manufacturer's protocol. Purified DNA extracts were quantified using dsDNA HS Assay kit Qubit fluorometer.

### Data generation, library preparation and sequencing

Each sample was PCR amplified targeting the 313bp fragment of the cytochrome oxidase I gene (COI) (mICOIintF: `GGWACWGGWTGAACWGTWTAYCCYCC` [25], matched to jgHCO2198: `TAIACYTCIGGRTGICCRAARAAYCA`; [26]). Each sample was amplified in triplicate and subsequently pooled to reduce biases in individual PCRs. A single step PCR protocol was used containing indexed primers with 8 bp oligo tags differing in at least 3 bases. A variable number (2, 3 or 4) of degenerate bases (N's) were added to the beginning of each primer to increase nucleotide diversity for sequencing. PCR reactions were carried out in 25 μL volumes containing 12.5 μL of MyFi mix (Bioline), 1 μL of each forward and reverse primer (0.5 μM), 2 μL of DNA template (0.5–10 ng/μL) and 8.5 μL of molecular grade $H_2O$. The

thermocycle condition for the PCR was 10 min at 94˚C; 35 cycles at 94 ˚C for 1 min, 45 ˚C for 1 min and 72 ˚C for 1 min; and a final elongation at 72 ˚C for five minutes. The quality of all amplifications was assessed through electrophoresis, running the PCR products on a 1% Sodium borate (1X SB) gel stained with gel red (Biotium). All PCR products were purified using magnetic beads (0.8x, Omega Mag-Bind) before all samples were pooled in equimolar amounts and normalized to 45 μL containing 3 μg of total purified PCR product. Along with the samples, one positive (*Astatotilapia burtoni*) and one negative (purified water) control was amplified in each plate and sequenced.

The Illumina library was constructed from 3 μg of total DNA using the NextFlex PCR-free library preparation kit following the manufacturer's instructions. The library was quantified by qPCR using NEBNext Library Quant Kit for Illumina, adjusted to a final molarity of 15 pM and with a 10% PhiX control, was sequenced on an Illumina MiSeq platform using v2 chemistry (2 x 250 bp paired-end) at the University of Hull.

## Bioinformatic pipeline

The sequence reads were analysed using the OBITools software [27]. FastQC was used to assess the quality of the reads and trimmed accordingly based on a minimum quality threshold of 28 using *obicut*. Pair-end reads were aligned using *illuminapairend* and alignments with a quality score <40 were discarded. The aligned dataset was demultiplexed using *ngsfilter*. The aligned reads were further filtered for length 300–320 bp (*obigrep*) and reads containing ambiguous bases were removed. The reads were then dereplicated using obiuniq and a chimera removal step was performed using the uchime-denovo algorithm implemented in vsearch [28]. Molecular Operational Taxonomic Unit (MOTU) clustering was carried out using Swarm (d value of 13) [29]. A reference COI database was generated by in silico PCR against the R134 release of the EMBL-EBI database using ecoPCR [30], and taxonomic assignment for each MOTU was performed using the *ecotag* algorithm, which implements a conservative lowest common ancestor approach.

## Statistical analysis

To determine adequate sampling of the diet profiles for each *Nephrops* collection site, we examined species diversity (presence-absence) of the gut contents using sample-based rarefaction and extrapolation sampling curves (iNEXT, [31]). The rarefaction curves were extrapolated for each collection site to 200 samples and the total species richness ($S_{est}$) for each site was estimated [32]. Rarefaction curves were used to determine the percentage of $S_{est}$ sampled for each site by dividing the cumulative number of expected species ($S_{est}$) by the estimated total species richness of each site (total $S_{est}$). We calculated indices of the relative frequency of occurrence as the occurrence per gut (O/G) index (the number of occurrences of a diet item divided by the total number of gut samples). Alpha diversity was performed using a pairwise ANOVA of diversity measure (Shannon index) of MOTUs for each site (*vegan*, [33]). Analyses were performed in the statistical programming environment R v.4.0.2 [34].

We employed Pianka's Niche Overlap Index [35] to quantify the dietary overlap for each site. This method facilitated a comparative analysis of dietary trends across the East-Irish, East-West, and Irish-West collections.

## Network analysis

We examined the dietary differences between *Nephrops* collections by generating a quantitative bipartite network [36] implemented in R (*geomnet*, [37]), where individuals grouped by location were linked to prey groups. The network was weighted to visualise the proportional

contribution of each prey group to the diet of *Nephrops* at a given location. Furthermore, a unipartite network was generated to illustrate the dietary preference for MOTUs at lower taxonomic ranks (species, genus or family level) between specimens among locations. This analysis consisted of one set of nodes whereby two species can be connected through trophic interactions [38]. The network was directed from predator to prey and interactions were weighted using presence-absence abundance for each MOTU. The final visualisation of the unipartite network was performed using the Force Atlas algorithm in Gephi v0.9.2 [39].

## Results

Overall, a total number of 207 *Nephrops norvegicus* specimens collected from three locations were screened for gut contents using DNA metabarcoding from the North Sea (NS-East (n = 77) and NS-West (n = 68)) and Irish Sea (n = 63), including one positive (*Astatotilapia burtoni*) and one negative (purified water) control. Illumina sequencing produced a total of 18,646,302 paired-end reads. After quality filtering (paired-end assembly, quality, length filtering and dereplication) and removing 21,592 potential chimeras (0.9%), the final table consisted of 815 MOTUs (12,228,986 reads). However, the vast majority of reads (98.6%, 12,054,773) aligned to Nephropidae which we considered host contamination and was therefore removed from further analysis. Furthermore, taxa unlikely to form the basis of *Nephrops* diet were removed (37 MOTUs, 5.2% reads, e.g., terrestrial species: hominid, Canidae, Insecta). We considered data with reliable taxonomic assignments 85% and greater, with each MOTU having a minimum of three reads. This resulted in a final dataset consisting of 94 *N. norvegicus* specimens (North Sea, East: 44, West: 25; Irish Sea: 25) with 116,154 reads in 119 MOTUs.

Rarefaction extrapolation curves were used to assess sampling effort of *Nephrops* diet among collections using presence-absence data (Fig 1). Rarefaction curves failed to reach saturation for each group suggesting increased sampling effort and/or sequencing depth is desirable to adequately obtain sufficient MOTU coverage. The analysis of total species richness revealed 74%, 49% and 46% of the estimated total MOTU richness was observed in diet composition from NSE (*n* = 44), NS-West (*n* = 25) and Irish Sea (*n* = 25) respectively. Rarefaction

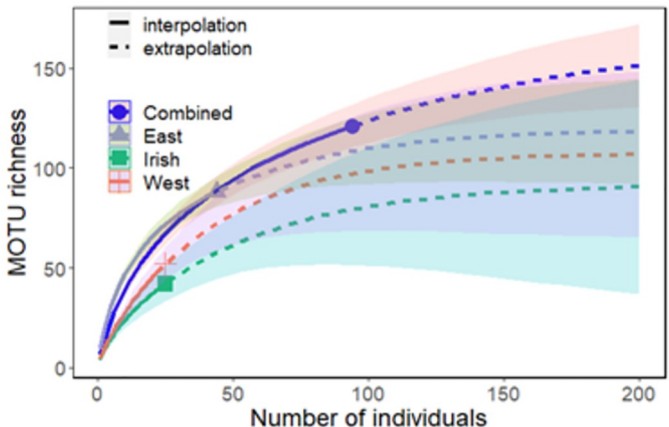

**Fig 1. Rarefaction and extrapolation curves for MOTU richness for diet sample estimates from three collections, with symmetric 95% confidence intervals based on [92].** Solid lines: rarefaction curves (observed data). Dashed lines: extrapolation curves. Shaded area for each solid line: 95% confidence interval for the expected rarefied MOTU richness. Shaded area for each dashed line: 95% confidence interval for the expected extrapolated class richness up to a sample size of 200. Points indicate observed rarefaction values for each collection.

**Table 1. Number of *N. norvegicus* specimens needed to detect various percentages of estimated total MOTU richness at three sites in the North Sea (East and West) and the Irish Sea.** *n* is the total number of samples analysed and Obs is the total MOTU richness observed for each site.

| Site | n | Obs | % Estimated total species richness | | | |
|---|---|---|---|---|---|---|
| | | | 80 | 90 | 95 | 99 |
| East | 44 | 74% | 62 | 101 | 130 | 158 |
| West | 25 | 49% | 77 | 119 | 147 | 175 |
| Irish | 25 | 46% | 84 | 131 | 163 | 194 |
| Combined | 94 | 80% | 93 | 148 | 186 | 224 |

curves indicated that an average of 74 and 117 individuals per site were needed to detect 80% and 90% of the estimated total MOTU species richness respectively (Table 1).

For alpha diversity analysis, we calculated the mean Shannon diversity and observed MOTUs found in the diet of *N. norvegicus* in each site (Fig 2). A one-way ANOVA analysis comparing diet diversity showed considerable differences across sites ($F = 5.17$, $p = 0.0067$). A Holm-Bonferroni corrected posthoc *t*-test showed the NS-East collection was notably different to both the NS-West and Irish Sea collections (paired *t*-test, p < 0.001). However, the observed number of MOTUs was higher in the NS-East (86) than the NS-West (50) and Irish Sea (43).

## Overall diet composition

The 119 MOTUs identified from the gastro-intestinal tract across *Nephrops* specimens consisted of 16 identified Phyla. The dietary overlap among *Nephrops* across different sites is notably high, as depicted in S1 Fig. Pairwise comparisons indicate substantial dietary overlap: 68% between East and Irish sites, 70% between East and West sites, and 87% between Irish and West sites. These high overlap percentages are likely attributable to the limited resolution of our sample collection. The bipartite plot demonstrates the broad associations of 22 prey groups between collections (Fig 3). The variety of prey sources was distributed over a diverse range of groups with Dinoflagellata as the most frequently occurring taxa in the gut content (average occurrence per gut, O/G = 0.44), followed by Chlorophyta (aO/G = 0.26), Holothuroidea (aO/G = 0.204), Malacostraca (aO/G = 0.203), Asteroidea (aO/G = 0.17), Bacillariophyta (aO/G = 0.16) and Nemertea (aO/G = 0.13) among others. While these groups appear abundant in the *Nephrops* diet, Fungi was found to be the most diverse with 28 MOTUs followed by Actinopterygii (12 MOTUs), Bacillariophyta (8 MOTUs), Malacostraca (5 MOTUs) and Ochrophyta (5 MOTUs) with the remaining groups showing between 1–4 MOTUs. We found

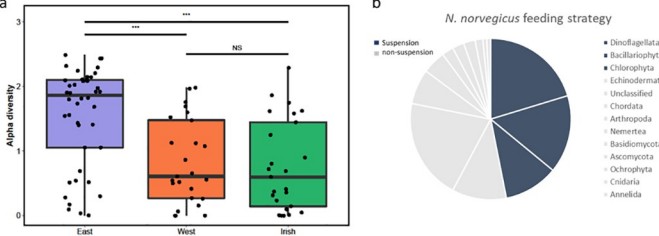

**Fig 2.** a) Box plots showing alpha diversity of *N. norvegicus* gut contents for each site, North Sea East, West and Irish Sea. The Shannon index was computed for all 94 specimens compared across locations. Bars above plots indicate significance *** p<0.001, NS non-significant. b) Pie chart illustrating *N. norvegicus* feeding strategy. Food likely consumed through suspension feeding are indicated by blue and non-suspension indicated by grey.

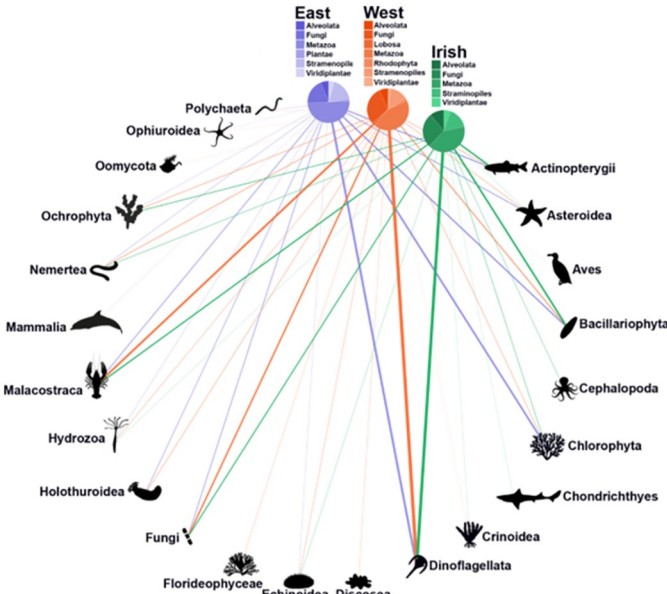

**Fig 3. Bipartite network including 94 *N. norvegicus* specimens (North Sea, East: 44, West: 25 and Irish Sea: 25) and their prey items (grouped into 22 taxonomic categories, including classes and phyla).** The relative proportion of each prey category consumed by each *N. norvegicus* group corresponds with the width of each interaction bar. The pie charts show the relative proportion of each taxonomic category within each group.

instances of some groups exclusively reported for the NS-East (e.g., Aves, Mammalia), NS-West (Discosea, Florideophyceae, Oomycota) and Irish Sea (Cephalopoda, Chondrichthyes).

## Prey composition

In our analysis of the dietary profiles of *Nephrops*, we focused on MOTUs identified at lower taxonomic ranks with >90% identity (58 MOTUs) to construct an empirical food web. This web consisted of 61 nodes and 94 weighted edges, representing predatory interactions based on presence-absence abundance (Fig 4). We detected 17 Fungi, 15 vertebrates, 15 invertebrates, and 11 algae/protists within the samples, where algae and protists are grouped together due to their overlapping characteristics. Key species within the network were identified through weighted degree centrality metrics, with algae/protists (*Suessiales* sp., O/G = 0.61; *Hematodinium* sp., O/G = 0.60; Dinophyceae sp., O/G = 0.11) emerging as the top MOTUs, (*Micromonas bravo*, O/G = 0.27; *Chloroparvula pacifica*, O/G = 0.15) followed by invertebrates (Common starfish, *Asterias rubens*, O/G = 0.22; Common sunstar, *Crossaster papposus* O/G = 0.10).

While our study detected some less likely dietary items, such as various Fungi and vertebrates, our primary focus is on the ecologically relevant prey items to better inform management decisions. For example, in the NS-East collection, we found a high prevalence of fish species like the common dragonet (Callionymus lyra, O/G = 0.02), European plaice (Pleuronectes platessa, O/G = 0.04), Atlantic herring (Clupea harengus, O/G = 0.02), and the common dab (Limanda limanda, O/G = 0.18). Other vertebrates included a seabird species, razorbill (*Alca torda*, O/G = 0.02), and two mammals, white-beaked dolphin (*Lagenorhynchus albirostris*, O/G = 0.02) and harbour porpoise (*Phocoena phocoena*, O/G = 0.04). Four invertebrates

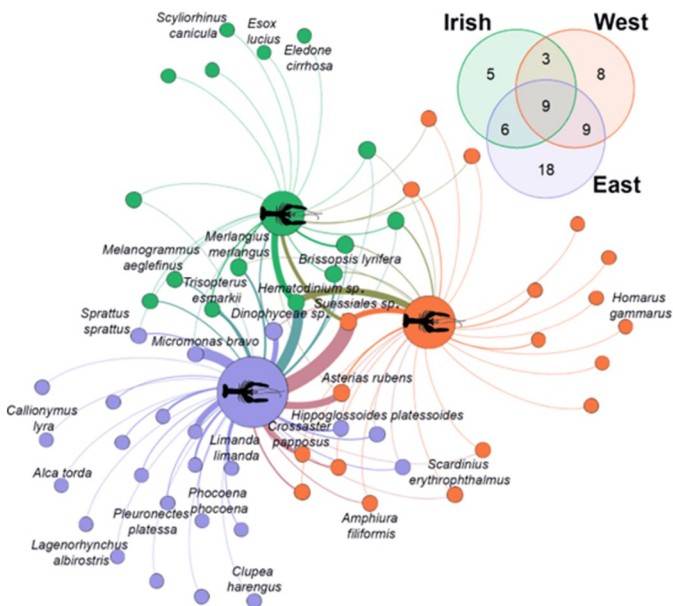

**Fig 4. Unipartite network illustrating species composition digested for *N. norvegicus* specimens collected at three sites.** Venn diagram shows the number of species level MOTUs per site.

detected were the black brittle star (*Ophiocomina nigra*, O/G = 0.02), two hydroids (*Nemertesia antennina* O/G = 0.02, *Leuckartiara octona*, O/G = 0.11) and a polychaete (*Pholoe pallida*, O/G = 0.04). By contrast, the NS-West was composed of eight unique MOTUs with two algae, two invertebrates, three Fungi and one amoeba detected. We detected an invertebrate hydroid MOTU in the Family Campanulariidae (O/G = 0.04) and the European lobster (*Homarus gammarus*, O/G = 0.04). Microalga species identified were the brown Forkweed alga (*Dictyota dichotoma*, O/G = 0.04) and red alga (*Ahnfeltia plicata*, O/G = 0.08). The Irish Sea showed the least unique MOTUs with two vertebrates (northern pike, *Esox lucius*, O/G = 0.04; small-spotted catshark, *Scyliorhinus canicula*, O/G = 0.04), one invertebrate (curled octopus, *Eledone cirrhosa*, O/G = 0.04) and two Fungi detected.

## Discussion

We used gut content metabarcoding of the commercially valuable *Nephrops norvegicus* to provide a detailed detection of specific food items to enhance traditional, broad-scale trophic assignments of prey. Our results indicate an opportunistic strategy that allows these generalist crustaceans to effectively utilize a wide range of food sources. These sources include macroalgae, phytoplankton (such as diatoms), fish, crustaceans, molluscs, echinoderms, nemerteans, polychaetes, mammals, fungi, and other taxa. The detection of these taxa in the *Nephrops* diet is based on the DNA extract gut contents, and while some items may be more prevalent or ecologically relevant than others, our findings present a comprehensive overview of the dietary profile for these crustaceans. In addition, a single snapshot of *Nephrops* diet revealed estimates of biodiversity and species distributions for a variety of food taxa in their native range, suggesting *Nephrops* could be used as a sentinel organism for monitoring local biodiversity. Last, we observed high rates of infection by the parasitic dinoflagellate *Hematodinium* in *Nephrops* populations from the North and Irish Sea, with over 50% of the sampled individuals found to be infected. *Hematodinium* is known to alter burrow-related behaviour in *Nephrops*, which may

have a detrimental impact on predation and fishing. Our sample collection and modest sequencing data lend further support for the promise of using DNA metabarcoding as a tool for measuring dietary profiles, biodiversity and monitoring pathogens in natural *Nephrops* populations that are important to commercial fisheries.

## Methodological considerations

DNA metabarcoding offers an enormous opportunity to observe greater taxonomic resolution to study the feeding ecology of organisms but a few drawbacks in our study require consideration to understand the dietary repertoire of *Nephrops*. First, isolating DNA from highly digested remains presents a methodological constraint for successful PCR amplification. Our initial objective was to include DNA metabarcoding data of the stomach contents of all *Nephrops* specimens, but we were unable to obtain sufficient PCR amplification and therefore we broadened our findings to gut contents which achieved greater PCR success. However, the presence of organic compounds in the stomach and gut of *Nephrops* (e.g., digestive enzymes, [40]) cause DNA damage which is one of the main contributors of PCR inhibition [41–43]. The variation in diet will affect the secretion of the gastric juices which play a significant role in PCR inhibition at different concentrations in the gut [44]. This can be highly problematic as high levels of PCR inhibition will be present in many genomic DNA extracts that negatively influence PCR amplification [45]. For more accurate identification and quantification of diet components, the following strategies could be considered to overcome DNA degradation and PCR inhibition: 1) DNA extraction protocols with tailored inhibition removal steps [46]; 2) incorporation of alternative proteins to enhance PCR amplification such as bovine serum albumin (BSA) or T4 gene 32 protein (gp32) [3, 45, 47] performing a DNA repair procedure on genomic DNA template to allow increased PCR amplification success [4, 48, 49] targeting a shorter fragment than the COI (313bp) such as the 18S rRNA V9 (~134 bp) to account for advanced digestion and DNA damage [50].

Second, the choice of primer is an important factor that influences the quality of PCR amplification and the desired taxonomic resolution. We used a versatile primer set that is known to be highly effective in amplifying COI across invertebrate phyla [25] with an expectation that high sequencing depth would recover an adequate characterisation of diet items. Consequently, we generated high sequencing depth with over 18 million sequencing reads but found that over 98% of the data was assigned to *Nephrops*. Although there are reports of cannibalism between conspecifics [51], we could not confidently distinguish between true cannibalistic events and host contamination in our genetic dataset. Therefore, we conservatively treated these sequences as host contamination and disregarded them in our analysis. Thus, there is considerable scope for improvement to identify species that are concealed by amplification bias. One solution to overcome host amplification is simply to target specific taxonomic groups such as vertebrates using a well-established 12S rRNA assay [18, 52] and the 18S rRNA assay [53]. This would greatly enhance the resolution of fish species detected to distinguish the proportion of discards that might contribute to the *Nephrops* diet. However, a multimarker assay would be needed to obtain a holistic view of eukaryotic diversity in the diet. Therefore, an alternative solution is the design of a blocking primer which can block amplification of host DNA in a complex sample [53, 54]. This can help preferentially bind non-host DNA through incorporating nucleotide mismatches for *Nephrops* in the primer with a C3 spacer on the 3′ end which blocks extension of the host PCR amplification [55, 56]. Development of a blocking primer assay for *Nephrops* will offer increased opportunity for species detection but it will require systematic PCR testing to determine the efficiency of blocking both non-target and target DNA. Nevertheless, our rarefaction estimates illustrate our COI assay recovered between

49% and 79% of prey diversity across sites for *Nephrops*, and additional improvements will reveal further fine-scale dietary information.

## Dietary profile of *Nephrops* norvegicus

Traditional diet composition analysis relies on visual observation methods of undigested remains and morphology-based stomach content analysis of *Nephrops* from the Mediterranean and Atlantic waters off the coast of Portugal show crustaceans and fish to be the main prey-groups [21]. However, this approach is labour-intensive, requires considerable taxonomic expertise, and the assessment of dietary items is often hampered by the variable rate of prey digestion in the gut [57], and lack of diagnostic features of digested and soft prey items, thereby underestimating the dietary assemblage. Stable carbon isotope composition of organic matter can trace the assimilation of nutrients present in animal tissue over a long period and Santana et al. [23] used stable isotope analysis of *Nephrops* collected in the west of Ireland and found suspended particulate organic matter and fish to be important diet components followed by plankton and invertebrate sources. Yet, this approach does not allow high-resolution analysis of species-specific diet composition.

Here we implemented DNA metabarcoding of the gastrointestinal tract using a COI primer set to characterise the diet of *Nephrops* from the North and Irish Sea. Overall, our molecular assessment of *Nephrops* gut content displays a broad omnivorous diet. This consisted of 16 Phyla and 35 Classes with 40 MOTUs (36%, ≥98% identity) identified to species level. We report the dominance of Dinoflagellata, Chlorophyta and Bacillariophyta which comprise an average of 49% of the dietary composition for *Nephrops* across sites and this may reflect a nutritional advantage consistent with their ability to suspension feed. Unclassified taxa alone made up an average of 8.7% of consumed prey and an average of 13% consisted of Echinodermata, Chordata and Arthropoda, while the remaining taxa (Ascomyata, Annelida, Basidiomycota, Cnidaria, Discosea, Mollusca, Nemertea, Ochrophyta, Oomycota, Rhodophyta) accounted for an average of 11%. Overall, we observed high dietary overlap among *Nephrops* across sites and this may reflect the common abundance of particulate organic matter and benthic organisms on which they feed. Similarly, morphological stomach content assessment of *Nephrops* from the Eastern and Western Mediterranean and adjacent Atlantic showed no significant differences between sites or seasons, which was explained by the great similarity of the bathyal fauna [21]. Nonetheless, we found instances of unique species in the diet of *Nephrops*, with 18 species exclusively found in the East of the North Sea (e.g., Actinopterygii, Aves, Mammalia), eight species in the West of the North Sea (Discosea, Florideophyceae, Oomycota) and five species found in the Irish Sea (Cephalopoda, Chondrichthyes).

*Nephrops* are benthic animals that burrow in soft sediment and emerge from their burrows to forage and seek mates. Fluctuations in food availability may present challenging conditions, particularly when their nutritional status is influenced by density-dependent factors. For example, in high density areas, competition for food may limit their scope for growth [20] and increased aggressive social behaviour could drive up the metabolic rate and thereby exhaust energy resources [58–60]. The significance of suspension feeding allows individuals to overcome challenging scenarios related to food availability, particularly for females during the breeding season [61], and avoiding aggressive encounters between male conspecifics [62]. It is suggested that suspension feeding is energetically efficient and can be more profitable for growth compared to active feeding given that 65–68% of the daily energy intake is achieved from suspension feeding [22, 58]. Our results reveal that nearly 60% of individuals (55/94) consume 50% or more taxa that are likely as a result of suspension feeding, a pattern that mirrors stable isotope analysis from Santana et al. [23]. This finding highlights the importance of

suspension feeding in *Nephrops* as a means of energy transfer within the ecosystem and its potential influence on local food web structure. The ability to utilize suspension feeding may also provide *Nephrops* with greater adaptability to changes in prey availability and enable efficient resource utilization. Additionally, the presence of parasitic or unintentionally consumed taxa in our results serves to underscore the diverse array of organisms encountered by *Nephrops* in their environment, offering valuable insights into the broader aspects of local biodiversity (Fig 2b). On the other hand, it is reported that females remain berried in normal environmental conditions to avoid predation during long breeding periods and depend on suspension feeding as an important strategy to survive starvation when restricted to burrows [22]. However, we found females to have similar dietary abundance as their male counterparts showing patterns of suspension and active feeding and this pattern is in line with Santana et al. [23], who revealed male and female *Nephrops* have remarkably similar dietary profiles. Therefore, it appears female *Nephrops* do not exclusively rely on suspension feeding and may experience occasional feeding excursions attracted by available food in close proximity to the burrow opening [61, 63]. Thus, the detection of active food foraging of prey such as invertebrates and fish along with other prey taxa shows the potential of *Nephrops* to utilise alternative, accessible, and highly nutritional prey to maximise their energy uptake.

More broadly, a combination of DNA metabarcoding and stable isotope analysis could significantly improve general data gathering to feed ecological models. The last broad scale description of diets in the North Sea ("Year of the Stomach") was carried out by a European consortium between 1981–1991 but, because of the labour-intensive nature of dietary identification at the time, was limited to identifying diets of a few commercial fish species and results were biased towards hard-bodied species [64]. Difficulties in identifying stomach contents often mean that ecological modellers use pooled taxa such as "plankton" or "heterotrophic benthos" in their diet matrices. Given the likely complexity within these groupings in terms of variations in numbers of component species in each classification, that are predated upon and their own trophic levels, this low resolution may limit the validity and utility of some ecological models [65, 66]. Metabarcoding may contribute to a route in developing much more robust diet matrices for ecological models.

## Sentinels of biodiversity

Biodiversity baselines play an important role in understanding the impact of multiple stressors such as climate change and anthropogenic pressures on biodiversity [67]. Monitoring programmes enable the observation of ecological temporal variability and enhance our capacity to manage species and ecosystems [68]. The past decade has experienced a surge in the development of DNA-based approaches to support rapid biodiversity assessments [18, 69], with recent work harnessing the 'natural sampler' ability of organisms to collect DNA from the environment. For example, Mariani et al. [70] utilised the water filtering efficiency of sponges to capture environmental DNA (eDNA) to recover highly informative biodiversity assemblages of vertebrate fauna from the Mediterranean Sea and Southern ocean. The concept of natural sampling has been further implemented in a range of different organisms to assess biodiversity using DNA metabarcoding of stomach contents from predatory or scavenging crustaceans [18, 71] to filter-feeding bivalve molluscs [53]. In the case of *Nephrops*, their potential as a natural sampler is enhanced by their wide distribution from Iceland to as far south of the Canary Islands with their range extending into the eastern Mediterranean Sea. *Nephrops* have a sustainable fishery and are relatively resilient to the effects of trawling in some areas of high fishing pressure with landings maintained at historically high levels for over 40 years [72]. This

means that these generalist predators and scavengers could be uniquely resourceful natural samplers in capturing benthic biodiversity.

The variation in food availability in the diet of *Nephrops* enhances the description of local biodiversity in the foraging area and helps build an inventory of species co-occurrence. We observed eight echinoderm species in the diet of *Nephrops* with overall greater occurrences of these species in the East of the North Sea (NS = East) than the NS-West and Irish Sea. Some of these echinoderms represent species with limited geographical distributions showing the centre of their abundances in the North and Irish Seas (e.g. the Spiny mudlark urchin *Brissopsis lyrifera*, brittlestar *Amphiura filiformis* and Black brittlestar *Ophiocomina nigra*). Other echinoderms detected have greater distribution ranges with abundances along the North Atlantic coasts of North America and Europe (Common starfish, *Asterias rubens*) to cosmopolitan species inhabiting cold temperate waters (Common sunstar, *Crossaster papposus*). Identification of these species is important to establish biodiversity baselines particularly for indicator species that help monitor ecosystem health. For example, Sea Star Wasting Disease (SSWD) is an ongoing disease epidemic that leads to behavioural changes, lesions, loss of turgor, limb autotomy, and death characterised by rapid degradation [73]. This caused mass mortality of major sea star populations along much of the west coast of North America resulting in the functional extinction of charismatic species (e.g., sunflower sea star, *Pycnopodia helianthoides*, [74]). Recently however, a SSWD-like outbreak has been documented and shown to be susceptible in the common sunstar (*Crossaster papposus*) in European waters from the Irish Sea and further research is needed to understand the geographical extent of the outbreak [75]. Although we report no detections of *S. papposus* in the Irish Sea, we observed an encouraging number of occurrences (n = 10) in the North Sea that illustrate *Nephrops* can be employed to monitor the presence of these keystone species. However, it is unclear whether the ingestion of echinoderms is a result of active predation or scavenging. For example, the Common starfish (*Asterias rubens*) is used as a biological indicator to assess the physical disturbance of bottom-trawl activity causing arm damage and leaving severed remains scattered along the seafloor [76]. While active predation on sea stars may occur, foraging the severed limbs can explain the occurrences in the diet of *Nephrops*, including the detection of other benthic species.

Fish comprise an important component of the *Nephrops* diet [21, 23]. The capture of certain fish may present little effort for *Nephrops* but it is argued that the consumption of fish is subsidised from discards of commercial fishing activity [23, 77]. However, it is unclear what species contribute to their consumption. We found twelve species of fish in the gut of *Nephrops* but with low occurrences across our samples. Most fish species consumed are demersal or benthopelagic that are commonly found in the Northeast Atlantic with several species also distributed throughout the Western Atlantic. These include haddock (*Melanogrammus aeglefinus*), whiting (*Merlangius merlangus*), Norway pout (*Trisopterus esmarkii*) and European sprat (*Sprattus sprattus*) which were detected in the North and Irish Seas, while Atlantic herring, (*Clupea harengus*), common dab (*Limanda limanda*), European plaice (*Pleuronectes platessa*) and American plaice (*Hippoglossoides platessoides*) were only detected in the North Sea. Many of these fish are commercially valuable and they constitute a substantial proportion of discards each year that negatively affects sustainable exploitation. For example, the mean discard rate for the European plaice between 2013–2017 accounted for a staggering 71% of the total catches from the Celtic Sea and Bristol Channel alone [78]. This high rate of discard mortality is intensified by the adverse impact of bottom trawling of non-target fish communities, and this is demonstrated from the whiting fishery as *Nephrops*-directed otter trawls accounted for 98% (1,030 tonnes) of discards from the Irish Sea in 2020 [79]. Therefore, a considerable proportion of fisheries discards contribute an important food source for these marine scavengers that offer favourable opportunities for growth. However, biodiversity estimates will be biased by seasonal

fisheries that alter the diversity that are available to *Nephrops*. Nevertheless, it is reasonable to consider biodiversity estimates of other naturally occurring species as a result of active foraging such as small demersal fish species with negligible discards or commercial importance (e.g., red bandfish, *Cepola macrophthalma*; dragonet, *Callionymus lyra*).

Our findings also revealed the presence of DNA from species that are likely to have been consumed through scavenging, such as jellyfish (*Leuckartiara octona*), octopus (*Eledone cirrhosa*), catshark (*Scyliorhinus canicular*), and dolphins (*Phocoena phocoena*, *Lagenorhynchus albirostris*). These species are known to inhabit the North Sea and Irish Sea, indicating that they are part of the native marine biodiversity in these regions. The detection of their DNA in the gut contents of *Nephrops norvegicus* underscores the generalist feeding behaviour of this crustacean, which includes opportunistic consumption of various food sources, such as carcasses and detritus containing tissue from these species.

This finding highlights the potential of *Nephrops norvegicus* as natural samplers for assessing local biodiversity, even for species that are not their primary prey. The presence of shark and dolphin DNA in their gut contents demonstrates the complex trophic interactions within marine ecosystems and provides valuable information on the role of *Nephrops* in these interactions. However, there is limited information available in the literature on similar patterns of crustaceans consuming or scavenging shark and dolphin tissue, emphasizing the novelty of our findings and the need for further research in this area. Future studies could explore the extent of scavenging behaviour in crustaceans and its implications for our understanding of marine food web dynamics and biodiversity assessments.

## Pathogen surveillance

Further to biodiversity monitoring, diet analyses performed using DNA metabarcoding with universal primers has an advantage of also detecting parasites as non-target identifications, demonstrating that parasites can be readily amplified from gut samples [53]. Our approach allowed a survey of parasitic dinoflagellates (e.g., *Hematodinium* sp.) that infect a growing number of crustacean genera globally, many of which are exploited as commercial fisheries [80]. They are considered the most significant known pathogen of *N. norvegicus* [81, 82]. We recovered the presence of *Hematodinium* in the gut of *Nephrops* in over 50% of individuals collected from the North and Irish Seas. The significance of these detections may indicate *Hematodinium*-infection which presents a negative impact on their behaviour, but the difference was marginal (overall prey observations: infected: 298, 52%; uninfected: 271, 48%). This implies infected individuals were exposed to more food resources possibly due to longer foraging durations. While the detection and presence of the parasitic dinoflagellate *Hematodinium* may represent a major stressor to *Nephrops*, further research is required to determine both the abundance of prey consumed as well as the progression and developmental stage of the parasite [83]. No attempt was made to confirm positive infection from our *Nephrops* collection as this finding represents a serendipitous result, but it is reasonable to expect positive detections are true infections based on the high prevalence of *Hematodinium*-infected *Nephrops* [83, 84]. Nevertheless, it is important to monitor the prevalence of this pathogen as it negatively affects swimming performance and burrowing behaviour for *Nephrops* individuals that experience a greater number of burrow departures and increased foraging time during illuminated periods compared to uninfected individuals [84]. This parasite is known to affect the hosts' behaviour indirectly through energy depletion [85] and an increased foraging time is essential to provide the nutritional requirements for the host and the parasite [84, 86]. We found a higher number of prey observations in *Hematodinium*-infected *Nephrops* compared to uninfected individuals, but infected individuals have a higher risk of predation and fishing

mortality, which in turn can have a pronounced impact on productivity and commercial catch quantity and value [83, 87].

## Implications for fisheries management

From a practical standpoint, the present results reveal important considerations for the assessment and management of these commercially valuable stocks. First, our gut content metabarcoding data clarifies complex food web structure by generating unparalleled resolution of trophic interactions, and this helps to overcome fragmented data with low resolution that has blurred existing ecosystem models of North Sea *Nephrops* to support an ecosystem approach to fisheries management [4]. Previous models were unable to acknowledge the significance of *Nephrops* feeding strategy through both suspension and active feeding for survival, revealed here with DNA based methods. Second, fisheries discards appear to play an important role in *Nephrops* nutrition as the fish species consumed are also among the highest recorded in discards from commercial fishing operations. The capture method used by commercial trawling (i.e., bottom trawling) will enable the rapid exploitation of fish species and indiscriminately catch non-target species which then become discarded and can eventually supply the *Nephrops* diet. However, while reducing bycatch and discarding remain conservation priorities, it is also crucial to understand and anticipate the potential consequences of reducing discards for species that have become reliant on them which may play an important structuring role for *Nephrops* as well as other taxa [88]. Last, it is difficult to ignore the strong pattern of *Hematodinium* sp. abundance among samples as this parasitic dinoflagellate is known to affect the burrow emergence behaviour of *Nephrops* [84] and negatively impacts the quality of the meat which can render them unmarketable and cause a significant economic loss to fisheries (estimated between GBP 2–4 million, [81, 83, 84, 89]). Further study is required to investigate the spatial and temporal prevalence patterns of *Hematodinium*-infected *Nephrops* as this will be a particularly important consideration for transferability [90] and the release of hatchery reared individuals in appropriate locations [91]. Given the added value of data from gut content metabarcoding we are in a position to radically improve the biological inference of *Nephrops* which can be implemented in ecosystem models to provide a greater understanding of trophic links with discards and help develop tailored management of this unique component of marine biodiversity.

## Conclusion

DNA metabarcoding offers enhanced taxonomic resolution in *Nephrops* dietary profiles beyond traditional morphology-based approaches and stable isotopes. The development of DNA metabarcoding-based inferences recently proposed using organisms to act as natural samplers for biodiversity assessments. We utilised the gut contents of *Nephrops* to characterise local biodiversity and estimate the prevalence of *Hematodinium* infected individuals. Our results show a strong dietary overlap in invertebrates, fish, algae, and other taxa from the North and Irish Seas and strengthen recent work indicating the significance of suspension feeding observed in *Nephrops*. Their generalist foraging behaviour allowed detection of indicator species used in routine environmental impact assessments and revealed the consumption of fish species associated with the high rate of discards. Importantly, DNA metabarcoding can complement, rather than fully replace, traditional gut content and stable isotope methodologies, as multi-trophic marker approaches provide a more holistic view of trophic dynamics. These patterns expand our understanding of *Nephrops* trophic ecology and offer interesting perspectives in methodological applications that indicate further avenues of research. Moreover, our findings underscore the potential importance of some "secondary" or unexpected

prey items in the *Nephrops* diet, suggesting that further research is needed to explore these dietary components in greater detail.

## Supporting information

**S1 Fig. Multidimensional Scaling (MDS) plot of diet variation of Nephrops norvegicus between sites, E: East (North Sea); Ir: Irish Sea; W: West (North Sea).**
(TIF)

## Acknowledgments

We thank the following for their help during the project; Edward Whittle of Whitby Seafoods for providing *Nephrops* samples; Mike Roach (SES Hull) for help with collection and dissection of animals used in this study; Robert Donnelly for technical assistance in the lab. Finally, we thank two anonymous reviewers for their constructive comments that improved the manuscript.

## Author Contributions

**Conceptualization:** Peter Shum, Magnus L. Johnson, Domino A. Joyce.

**Data curation:** Peter Shum.

**Formal analysis:** Peter Shum, Graham S. Sellers.

**Funding acquisition:** Domino A. Joyce.

**Investigation:** Peter Shum.

**Methodology:** Peter Shum, Janine Wäge-Recchioni.

**Resources:** Magnus L. Johnson, Domino A. Joyce.

**Writing – original draft:** Peter Shum.

**Writing – review & editing:** Peter Shum, Janine Wäge-Recchioni, Graham S. Sellers, Magnus L. Johnson, Domino A. Joyce.

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
