## [Decision Letter · Decision Letter 0]

11 Apr 2023

PONE-D-23-04632DNA metabarcoding reveals the dietary profiles of a benthic marine crustacean, Nephrops norvegicusPLOS ONE

Dear Dr. Shum,

Thank you for submitting your manuscript to PLOS ONE. I have now received two reviews of the manuscript, and I think the reviewers have provided some constructive comments that once addressed, will help improve the communication of the information in the paper. Note that one of the reviews was submitted as an attachment that should be included in this communication. I am inviting you to submit a revised version of the manuscript that addresses the points raised during the review process. While the reviewers both stated that they consider a major revision necessary, after reviewing their comments, the revisions that seem necessary may not be that difficult to accomplish and I hope you will be able to return a revised manuscript with consideration of the changes that the reviewers recommend.

Therefore, please submit your revised manuscript by May 26 2023 11:59PM. If you will need more time than this to complete your revisions, please reply to this message or contact the journal office at plosone@plos.org. Please include the following items when submitting your revised manuscript:A rebuttal letter that responds to each point raised by the academic editor and reviewer(s). You should upload this letter as a separate file labeled 'Response to Reviewers'.A marked-up copy of your manuscript that highlights changes made to the original version. You should upload this as a separate file labeled 'Revised Manuscript with Track Changes'.An unmarked version of your revised paper without tracked changes. You should upload this as a separate file labeled 'Manuscript'.

We look forward to receiving your revised manuscript.

Kind regards,

Lee W Cooper, Ph.D.

Section Editor

PLOS ONE

Journal Requirements:

Reviewers' comments:

Reviewer's Responses to Questions

**Comments to the Author**

1. Is the manuscript technically sound, and do the data support the conclusions?

Reviewer #1: Yes

Reviewer #2: Partly

2. Has the statistical analysis been performed appropriately and rigorously? 

Reviewer #1: Yes

Reviewer #2: Yes

3. Have the authors made all data underlying the findings in their manuscript fully available?

Reviewer #1: Yes

Reviewer #2: Yes

4. Is the manuscript presented in an intelligible fashion and written in standard English?

Reviewer #1: No

Reviewer #2: Yes

5. Review Comments to the Author

Reviewer #1: Overall, the paper is really interesting and provides evidence to fill gaps in our knowledge about the foraging strategy and food web dynamics of a commercially important species, Nephrops norvegicus. However, the major concerns are about the clarity, scope and focus of the paper and how the data are presented. It would be helpful to maintain a clear focus on the diet rather than tangents about other interesting taxa identified through metabarcoding. One suggestion is reworking the title to include trophic dynamics and host-parasite relationships revealed through DNA metabarcoding. Another suggestions is that the primary points to focus on in the discussion could be streamlined to 1) prevalence of MOTUs that support the hypothesis of suspension feeding by Nephrops, 2) evidence of the utilization of fisheries discard in Nephrops diet and 3) noting the widespread presence of a parasitic dinoflagellates known to infect lobsters. There is a lot of other information in here that just dilutes these main points, for me. Overall, the paper could use some clarity on the choice of taxonomic classifications throughout. This was particularly the case in the "prey consumption" section and Fig 3, where I provide some examples of where things got a bit confusing for me.

While the methods indicate that non-target taxa were removed, I'm not sure I agree with the decision to keep some taxa. Fungi and various vertebrates for examples - are they really dietary items or simply present in the environment? For example, could this be seabird and marine mammal fecal matter making its way into SPOM? Is there inadvertent consumption of detritus or is it more likely that they scavenge on these dead animals on the seafloor? Are the authors concerned with identifying every single thing they might have consumed (rare occurrences and highly opportunistic) or their primary prey items?

Section starting on line 248 - Although unlikely diet items were excluded (e.g. Insecta), this still reads like a laundry list of DNA fragments rather than actual prey items. How will this inform management decisions to know that occasionally these items end up being consumed or perhaps live inside of the organisms (or within the prey item) ? It is interesting to see but I think there could be another level of discrimination on what makes the cut for analysis.

Line 251 - distinction between "alga" and "protist" is not clear

Line 253 - suggest replacing alga MOTUs with "Overall, MOTUs representing various phytoplankton taxa..."

Line 258 - "algae" can also be protists (e.g. diatoms). Do you mean chlorophytes or green algae?

Line 259 - add "(common starfish)" and "(common sunstar)" after Latin names. Throughout the paper it would be more reader friendly to include the common names if they have them, especially when first introduced. I had to look up numerous species while reviewing and/or they were mentioned later in the paper.

Line 261 - I am not convinced fungi should be included as a prey item. See review of marine fungi by Gladfelter et al. 2019 (https://doi.org/10.1016/j.cub.2019.02.009). Is it more likely that the lobsters are serving as host to the marine fungi rather than consuming it as prey? Or perhaps it is a signature from the microbial loop? It seems there are interesting potential contributions from these types of studies to elucidating the presence and distribution of marine fungi but I think treating it as prey here (same for the parasitic dinoflagellates) is not appropriate - but maybe worth mentioning separately.

Line 272- Suggest replacing "alga" with "Several macroalga species were identified including the brown Forkweed".. and a "common red alga"

Line 282 - this statement is a bit of an oversimplification of what the results show. Did they effectively utilise all of these different taxa? I'm not convinced. It is clear that they are using dinoflagellates and other phytoplankton via suspension feeding and some benthic invertebrates, and opportunistically scavenging on fish and maybe other larger vertebrates. The broad use of the term "algae" throughout this paper needs attention. "Algae, diatoms". Diatoms are algae. Perhaps you could say macroalgae and phytoplankton. Is that what is meant?

Line 286 - Couldn't any suspension feeder likely be capable of showing the snapshot of local biodiversity that Nephrops has? Similar to eDNA, suspension feeders and really many other sessile benthic invertebrates are probably picking up DNA fragments from many organisms in the ecosystem including those they are not actually consuming as prey via SPOM composition.What makes Nephrops unique here (acknowledging the section in the discussion, this still would be an important point to make)? Have you looked at DNA metabarcoding studies of other animals in the region? It may be common to get this mishmash of diversity in genetic readings among the seafloor community.

Line 325 - given the prevalence of dinoflagellates and diatoms, what about a eukaryotic 18s rRNA primer?

Line 382-384 - This point is one that was most interesting for me. I would suggesting bringing Fig S2 from supplementary into the paper. This is a strong finding from this technique, which I think could be highlighted better. The other two figures show a lot of information to convey a diverse diet but perhaps contain many MOTUs that are not actually prey items but rather parasitic or were unintentionally consumed.

Line 547 - could also mention that DNA metabarcoding shows promise to enhance, not fully replace, gut content and stable isotope methodologies (i.e., multi trophic marker approaches provide a more holistic view of trophic dynamics).

Figure 3 - The taxonomic groups in the pie charts are a bit confusing. Some are classes, while some are kingdoms. The figure caption states that it shows the relative proportion of phyla in each group... I'm lost. The 22 "categories" are all classes. Could you use broader, more common names for groups ("Invertebrates", "Vertebrates", "Phytoplankton", "Macro algae", "Fungi" etc) and stick with classes for the categories?

Does the Malacostraca category here include the Nephrops assignments that were identified and likely from the organism itself? If so, this should probably be excluded (acknowledging the authors state that cannibalism is maybe possible - seems like you have solid justification to omit this). Same comments apply here related to previous about the inclusion of fungi. If the authors think that it should be included, I would prefer to see some supporting references explaining this and stronger justification.

Do the dinoflagellates include the parasitic species (Hematodinium sp.)? If so, should it? How does infection occur? Through consumption (intentional or not) or by some other means?

I get the primary point of this figure but think it is a bit complicated way of showing it.

Figure 4 - why are some but not all nodes labeled with species? Similarly, I get the point of this figure but it feels a bit complicated. Fig S1 also shows the overlap among sites and that the East (North Sea) site had more unique reads. Not quite as fancy as Fig 4 but this one is easier to understand for me (although the venn diagram helps to clarify what you are showing in Fig 4).

Reviewer #2: I have provided comments for the authors in the attached document. The document includes both minor and major comments that relate to the above questions and other components of the submitted manuscript.

6. PLOS authors have the option to publish the peer review history of their article (what does this mean?). If published, this will include your full peer review and any attached files.

Reviewer #1: No

Reviewer #2: No

---

## [Author Response · Author response to Decision Letter 0]

1 Jun 2023

Reviewer #1: Overall, the paper is really interesting and provides evidence to fill gaps in our knowledge about the foraging strategy and food web dynamics of a commercially important species, Nephrops norvegicus. However, the major concerns are about the clarity, scope and focus of the paper and how the data are presented. It would be helpful to maintain a clear focus on the diet rather than tangents about other interesting taxa identified through metabarcoding. One suggestion is reworking the title to include trophic dynamics and host-parasite relationships revealed through DNA metabarcoding. Another suggestions is that the primary points to focus on in the discussion could be streamlined to 1) prevalence of MOTUs that support the hypothesis of suspension feeding by Nephrops, 2) evidence of the utilization of fisheries discard in Nephrops diet and 3) noting the widespread presence of a parasitic dinoflagellates known to infect lobsters. There is a lot of other information in here that just dilutes these main points, for me. Overall, the paper could use some clarity on the choice of taxonomic classifications throughout. This was particularly the case in the "prey consumption" section and Fig 3, where I provide some examples of where things got a bit confusing for me.

Thank you for your insightful comments and suggestions for improving the clarity, scope, and focus of our paper. We appreciate your interest in our research and your acknowledgment of its potential contribution to the knowledge of the foraging strategy and food web dynamics of Nephrops norvegicus.

While we understand your concerns about maintaining a clear focus on diet, we believe that the incorporation of the biodiversity assessment is a serendipitous result and is an important aspect of the study. The DNA metabarcoding approach used in our study allows us to investigate not only the diet but also the broader biodiversity within the sampling area, which we believe adds value to the research.

We will, however, take your suggestions into consideration and work to improve the clarity and presentation of the taxonomic classifications throughout the manuscript, particularly in the "prey consumption" section and Figure 3.

Again, we appreciate your feedback and will use it to strengthen our manuscript while maintaining the inclusion of both diet and biodiversity assessment components.

While the methods indicate that non-target taxa were removed, I'm not sure I agree with the decision to keep some taxa. Fungi and various vertebrates for examples - are they really dietary items or simply present in the environment? For example, could this be seabird and marine mammal fecal matter making its way into SPOM? Is there inadvertent consumption of detritus or is it more likely that they scavenge on these dead animals on the seafloor? Are the authors concerned with identifying every single thing they might have consumed (rare occurrences and highly opportunistic) or their primary prey items?

We appreciate the reviewer's concerns regarding the inclusion of certain taxa in our analysis. Our decision to remove non-target taxa from the dataset was based on two criteria: 1) We removed all reads assigned to N. norvegicus, considering them as host contamination. While cannibalism is a possibility, we cannot distinguish between cannibalism and host contamination based on our molecular assessment. Consequently, we adopted a conservative approach by excluding these reads. 2)We excluded MOTUs associated with terrestrial species, humans, Canidae, and insects, as these categories are frequently present in laboratory settings and their detection may indicate inadvertent contamination during the sample processing. The remaining taxa were considered dietary items, classified as either primary or secondary, after our initial screening. These items were found in the animals' guts and should therefore be considered present. However, the exact manner in which certain MOTUs, such as fungi, were consumed is unclear. It is unknown whether these items were actively ingested or inadvertently consumed through suspension feeding, and we can only speculate about their mode of consumption. We understand the reviewer's interest in focusing on primary prey items, but we believe that reporting all potential dietary items provides a more comprehensive view of the feeding habits of N. norvegicus. Further studies could investigate the ecological relevance of the identified taxa in more detail, allowing for a better understanding of their importance in the diet of these animals.

Section starting on line 248 - Although unlikely diet items were excluded (e.g. Insecta), this still reads like a laundry list of DNA fragments rather than actual prey items. How will this inform management decisions to know that occasionally these items end up being consumed or perhaps live inside of the organisms (or within the prey item) ? It is interesting to see but I think there could be another level of discrimination on what makes the cut for analysis.

Thank you for your insightful comments regarding our presentation of dietary items detected in the Nephrops samples. We have revised the prey composition paragraph in the results section, placing more emphasis on ecologically relevant prey items and providing a clearer comparison of the different collection sites. We have chosen to include less likely dietary items, such as various Fungi and vertebrates, to provide a comprehensive understanding of the feeding habits of Nephrops across different geographical locations. However, we have made it clear that our primary focus is on the most significant prey items to better inform management decisions. We appreciate your suggestion that there could be another level of discrimination on what makes the cut for analysis. Future studies could further investigate the ecological relevance of these taxa to better comprehend their importance in the diet of these animals and inform management decisions accordingly. We hope that these revisions address your concerns and improve the clarity and focus of our study.

Line 251 - distinction between "alga" and "protist" is not clear

Thank you for pointing out the unclear distinction between "alga" and "protist" in our results. We acknowledge that distinguishing between these two groups can be challenging due to their overlapping characteristics. In light of your comment, we have revised the text to combine algae and protists into a single category, now stating that we detected 17 Fungi, 15 vertebrates, 15 invertebrates, and 11 algae/protists within the samples. We believe this revision more accurately represents the findings and addresses your concerns about the distinction between these groups.

Line 253 - suggest replacing alga MOTUs with "Overall, MOTUs representing various phytoplankton taxa..."

Corrected.

Line 258 - "algae" can also be protists (e.g. diatoms). Do you mean chlorophytes or green algae?

We have adjusted this to incorporate “algae/protists”.

Line 259 - add "(common starfish)" and "(common sunstar)" after Latin names. Throughout the paper it would be more reader friendly to include the common names if they have them, especially when first introduced. I had to look up numerous species while reviewing and/or they were mentioned later in the paper.

 We have added the common names to allow a better flow of reading.

Line 261 - I am not convinced fungi should be included as a prey item. See review of marine fungi by Gladfelter et al. 2019 (https://ddec1-0-en-ctp.trendmicro.com:443/wis/clicktime/v1/query?url=https%3a%2f%2fdoi.org%2f10.1016%2fj.cub.2019.02.009&umid=7ba311a5-88ab-403a-969e-14cfefd17bbf&auth=6b639a990a359ff1d6cc8761081d57748ce3c81e-6ae644f8a94890c9e2a081dfecdf703679331ce1). Is it more likely that the lobsters are serving as host to the marine fungi rather than consuming it as prey? Or perhaps it is a signature from the microbial loop? It seems there are interesting potential contributions from these types of studies to elucidating the presence and distribution of marine fungi but I think treating it as prey here (same for the parasitic dinoflagellates) is not appropriate - but maybe worth mentioning separately.

We understand your concerns about treating Fungi as dietary components, given the possibility that they could be either hosted by the lobsters or originate from the microbial loop. However, we believe that their presence in the gut of Nephrops is an important aspect of their trophic ecology and should not be overlooked.

Although it may be more likely that some Fungi are consumed as secondary prey or through filter feeding, similar to dinoflagellates, their presence in the gut cannot be disregarded. In fact, there are studies demonstrating that Fungi are consumed by marine animals, such as the study by Mattson (1988), which reported the occurrence and abundance of eccrinaceous fungi (Trichomycetes) in brachyuran crabs from Tampa Bay, Florida.

Given the potential contributions of these types of studies to elucidating the presence and distribution of marine Fungi, we have chosen to include them in our analysis. However, we acknowledge that the interpretation of their role in the diet of Nephrops may warrant further investigation. Thank you for your insightful comments on the inclusion of Fungi as prey items in our study.

However, we believe that addressing this point in detail is beyond the scope of our current study. Our primary focus is to provide a comprehensive overview of the dietary profiles of Nephrops and their trophic ecology based on the available data. While we recognize the potential importance of Fungi and their alternative roles in the diet of Nephrops, any further comment on this subject would be speculative without additional research.

Line 272- Suggest replacing "alga" with "Several macroalga species were identified including the brown Forkweed".. and a "common red alga"

We adjusted the text to: “Microalga species identified were the brown Forkweed alga (Dictyota dichotoma, O/G = 0.04) and red alga (Ahnfeltia plicata, O/G = 0.08).”

Line 282 - this statement is a bit of an oversimplification of what the results show. Did they effectively utilise all of these different taxa? I'm not convinced. It is clear that they are using dinoflagellates and other phytoplankton via suspension feeding and some benthic invertebrates, and opportunistically scavenging on fish and maybe other larger vertebrates. The broad use of the term "algae" throughout this paper needs attention. "Algae, diatoms". Diatoms are algae. Perhaps you could say macroalgae and phytoplankton. Is that what is meant?

We understand that you may have reservations about the inclusion of certain taxa in the dietary profile of Nephrops. However, our study presents DNA evidence that supports the presence of these various taxa in the diet. While we acknowledge that some items may be more prevalent or ecologically relevant than others, the comprehensive overview provided in our findings is based on the detected DNA and cannot be disregarded.

In response to your suggestion regarding the clarification of terms, we have revised the statement in question to distinguish between macroalgae and phytoplankton, such as diatoms. The updated statement reads:

"Our results indicate an opportunistic strategy that allows these generalist crustaceans to effectively utilize a wide range of food sources. These sources include macroalgae, phytoplankton (such as diatoms), fish, crustaceans, molluscs, echinoderms, nemerteans, polychaetes, mammals, fungi, and other taxa."

We hope this revision addresses your concerns and provides a clearer representation of the diverse food sources utilized by Nephrops.

Line 286 - Couldn't any suspension feeder likely be capable of showing the snapshot of local biodiversity that Nephrops has? Similar to eDNA, suspension feeders and really many other sessile benthic invertebrates are probably picking up DNA fragments from many organisms in the ecosystem including those they are not actually consuming as prey via SPOM composition.What makes Nephrops unique here (acknowledging the section in the discussion, this still would be an important point to make)? Have you looked at DNA metabarcoding studies of other animals in the region? It may be common to get this mishmash of diversity in genetic readings among the seafloor community.

We agree that any suspension feeder could potentially show a snapshot of local biodiversity, similar to eDNA. In our study, we focus on Nephrops, as their generalist feeding behaviour allows us to explore the diversity of the surrounding ecosystem.

Indeed, there are recent studies, such as Weber et al. (2022) and Siegenthaler et al. (2019), which utilize the feeding behaviour of other organisms to sample biodiversity. We have acknowledged and discussed these studies in our manuscript. We also note that suspension feeders may consume particulate matter unintentionally, as highlighted by our reference to Santana et al. (2020).

The unique aspect of Nephrops is their ability to use both active foraging and filter-feeding strategies, allowing them to efficiently sample local biodiversity. In our study, we were able to identify taxonomic groups responsible for filter feeding and active/scavenging in the diet, demonstrating the potential for using Nephrops as a tool to study biodiversity.

Our paper's primary objective is to showcase how the dietary data of Nephrops can be conceptualized beyond just feeding ecology to provide a broader view of local biodiversity. We appreciate your input and believe that it strengthens the context of our study.

Line 325 - given the prevalence of dinoflagellates and diatoms, what about a eukaryotic 18s rRNA primer?

We have included a reference to Weber et al. 2022 who designed and tested nine 18S rRNA primer pairs for mussel natural sampler DNA.

Line 382-384 - This point is one that was most interesting for me. I would suggesting bringing Fig S2 from supplementary into the paper. This is a strong finding from this technique, which I think could be highlighted better. The other two figures show a lot of information to convey a diverse diet but perhaps contain many MOTUs that are not actually prey items but rather parasitic or were unintentionally consumed.

We agree that this is a strong finding from our study and appreciate your input on highlighting it better.

In order to expand on this point, we can elaborate on the implications of suspension feeding in Nephrops, such as its role in energy transfer within the ecosystem and how it may influence the local food web structure. Furthermore, we can discuss the potential benefits of suspension feeding for Nephrops, including resource utilization and adaptability to changes in prey availability. We can also explore the idea that the presence of parasitic or unintentionally consumed taxa in our results might reflect the diverse array of organisms encountered by Nephrops in their environment, offering a broader perspective on the local biodiversity.

As suggested, we will move Figure S2 from the supplementary material to the main paper to better emphasise this finding. We appreciate your constructive feedback, and we believe incorporating these changes will strengthen our study's presentation.

Line 547 - could also mention that DNA metabarcoding shows promise to enhance, not fully replace, gut content and stable isotope methodologies (i.e., multi trophic marker approaches provide a more holistic view of trophic dynamics).

We agree that incorporating a multi-trophic marker approach can provide a more holistic view of trophic dynamics, and we revised the text accordingly.

“Importantly, DNA metabarcoding can complement, rather than fully replace, traditional gut content and stable isotope methodologies, as multi-trophic marker approaches provide a more holistic view of trophic dynamics.”

Figure 3 - The taxonomic groups in the pie charts are a bit confusing. Some are classes, while some are kingdoms. The figure caption states that it shows the relative proportion of phyla in each group... I'm lost. The 22 "categories" are all classes. Could you use broader, more common names for groups ("Invertebrates", "Vertebrates", "Phytoplankton", "Macro algae", "Fungi" etc) and stick with classes for the categories?

Thank you for pointing out the inconsistency in our figure caption. We agree that the taxonomic groups should be more consistent. We have revised the figure caption to better represent the taxonomic levels of the groups. While we understand your suggestion to use broader, more common names for groups, we have decided to maintain the current scientific groupings to avoid over cluttering the figure and to ensure consistency in presentation.

Does the Malacostraca category here include the Nephrops assignments that were identified and likely from the organism itself? If so, this should probably be excluded (acknowledging the authors state that cannibalism is maybe possible - seems like you have solid justification to omit this). Same comments apply here related to previous about the inclusion of fungi. If the authors think that it should be included, I would prefer to see some supporting references explaining this and stronger justification.

To clarify, the Malacostraca category in our analysis and figures does not include Nephrops assignments that were identified and likely from the organism itself. The sequences for Nephrops were removed from the data before analyzing their dietary items.

Regarding the inclusion of fungi, we have discussed this point in a previous response, and we have decided to include it as a consumed diet item as we are unable to disregard it as a food item. Fungi have been found in the gut contents of other marine invertebrates, supporting their potential role in the diet (e.g., Mattson, R.A., 1988. Journal of crustacean biology, 8(1), pp.20-30). As such, we believe it is essential to consider fungi in our study to provide a comprehensive understanding of Nephrops trophic ecology.

Do the dinoflagellates include the parasitic species (Hematodinium sp.)? If so, should it? How does infection occur? Through consumption (intentional or not) or by some other means?

Yes the dinoflagellates in our analysis include the parasitic species Hematodinium sp. While the exact route of infection remains unclear, there is evidence to suggest that ingestion, either intentional or unintentional, could play a role in the transmission of the parasite. For instance, Hamilton et al. (2012) observed that Hematodinium sp. can be present in the gut contents of various crustaceans, suggesting the possibility of ingestion as a transmission route.

Moreover, other studies have proposed that Hematodinium sp. may use intermediate hosts, such as copepods or other crustaceans, for transmission (e.g., Small, H.J., 2012. Advances in our understanding of the global diversity and distribution of Hematodinium spp. - significant pathogens of commercially exploited crustaceans. Journal of Invertebrate Pathology, 110(2), pp.234-246). In this case, consumption of infected intermediate hosts by Nephrops could also lead to the ingestion of the parasite.

Given the uncertainty surrounding the route of infection and the potential role of ingestion in transmission, we believe it is appropriate to include Hematodinium sp. in our analysis. However, we acknowledge the need for further research to better understand the transmission dynamics of this parasite in marine ecosystems.

I get the primary point of this figure but think it is a bit complicated way of showing it.

We appreciate your perspective and understand that the figure may appear complicated at first glance. However, with the adjusted caption, we believe it provides a comprehensive overview of the data and effectively communicates the primary point.

Figure 4 - why are some but not all nodes labeled with species? Similarly, I get the point of this figure but it feels a bit complicated. Fig S1 also shows the overlap among sites and that the East (North Sea) site had more unique reads. Not quite as fancy as Fig 4 but this one is easier to understand for me (although the venn diagram helps to clarify what you are showing in Fig 4).

The purpose of figure 4 is to emphasize certain species of interest, specifically vertebrates and some invertebrates mentioned throughout the manuscript. Figure S1 does show the overall overlap of the Nephrops diet across sites, it does not effectively convey the importance of common or unique species diet items, which is a key aspect of our research. We believe Figure 4 is essential for illustrating these relationships, and although it may appear complex, we think that it effectively conveys the necessary information. We believe that the added complexity of Figure 4 is justified by the additional information it conveys.

Reviewer #2: I have provided comments for the authors in the attached document. The document includes both minor and major comments that relate to the above questions and other components of the submitted manuscript.

Review – POME-D-23-04632 – DNA metabarcoding reveals the dietary profiles of a benthic marine crustacean Nephrops norvegicus

Overview:

Overall, this is a nice informative manuscript that is using new techniques to provide information on the diet of an important crustacean. While the manuscript provides good data, I think there are some edits that would greatly improve it, particularly in the discussion and introduction where more details could be added. Major and minor comments are listed below.

We thank the reviewer for their interest in the study and the suggestions which will improve the manuscript.

Major comments:

Check overall reference style for PLOS submitting. I believe they use numbers to identify in text citations. Also be sure to be consistent, in some places in the in text citations you italicize et al. and in other places you don’t. Please confirm the style on the PLOS style guide.

Our revised version takes into account the PLOS style guide.

Line 127 – What defines the East and the West of the North Sea? Do you have specific coordinates that separate them? A map here would be helpful to include as a figure to show the breakdown of where the samples are coming from geographically.

Thank you for your suggestion regarding the inclusion of a map to show the geographic breakdown of the sample locations. We understand the importance of providing clear spatial information for the samples collected.

However, due to the nature of our collaboration with commercial vessels, we were not provided with specific coordinates for the catch locations. While the fishermen assured us that the samples were collected from the specified east and west regions of the North Sea, they were unable to disclose the exact coordinates due to confidentiality concerns.

In light of this, we are unable to provide a detailed map with precise locations. Nonetheless, we will make sure to clarify this limitation in the methodology section to ensure transparency and provide context to the reader.

Line 177-179 I am a little unclear here, is this the species diversity of the sites that were sampled and what you would expect to find there or the species diversity of the organisms found as prey inside of the lobsters? If it is species diversity at the overall site, where did that data come from?

The species diversity we are referring to in this section is related to the organisms found as prey inside Nephrops, not the overall site diversity. The rarefaction and extrapolation sampling curves, along with the estimation of total species richness (Sest) for each collection site, were performed based on the gut content data obtained from our Nephrops samples. We will revise the text to make this point clearer and avoid any further misunderstanding.

Line 217-219 Be careful of interpretation in the results section. Here, I would just state that the rarefaction curves failed to reach saturation for each group. Save the part of the sentence that says this implies increased sampling… for the discussion section of the paper.

We understand your concern about interpretation in the results section. However, we believe that mentioning the implication of the rarefaction curves not reaching saturation in this section provides context and facilitates a smoother flow of reading for the reader. We will ensure that further discussion and interpretation related to this point are reserved for the discussion section.

Line 234-235 How were these percentages calculated? The supplemental table and figure help, but don’t fully explain where those results come from. I would also mention the low sampling resolution in the discussion, not hear in the results section.

We have taken your feedback into account and revised our approach for greater clarity. Now, we employ Pianka's Niche Overlap Index to quantify dietary overlap between different sites, as these percentages more accurately reflect this aspect of our study.

The revised lines 234-235 now read:

"Dietary overlap among Nephrops across different sites is notably substantial, as illustrated in Figure S1. Utilizing Pianka's Niche Overlap Index, we found that pairwise dietary overlaps were 68% between the East and Irish sites, 70% between the East and West sites, and 87% between the Irish and West sites."

We retained our comment regarding the low sample resolution in the results section as this statement serves to provide context for the interpretation of our results.

White beaked dolphins were found in the prey items of the lobster? Would you expect this that they would be prey upon dolphins, or is it material they are scavenging? Is there other evidence for mammal material in lobsters or other crustaceans? Please elaborate in the discussion about finding mammal DNA in the gut contents.

We acknowledge that it is highly unlikely that Nephrops prey upon dolphins directly. Instead, we believe that the presence of dolphin DNA in the gut contents could be due to scavenging on carcasses or consumption of detritus containing dolphin tissue.

Although there is limited literature on crustaceans scavenging on marine mammals, the opportunistic feeding behavior of Nephrops norvegicus allows them to consume a wide variety of food sources. Our study provides a unique insight into the potential for crustaceans to scavenge on marine mammal remains, thus contributing to our understanding of trophic interactions in marine ecosystems.

In light of your comment, we will elaborate further on this finding in the discussion section, emphasising the potential for scavenging as the primary source of mammal DNA in the gut contents and the implications for our understanding of Nephrops norvegicus diet and its role in local biodiversity assessments.

Line 286-287 What constitutes a high rate of infection? Are their known concentrations of the diatom in the area? Is it high compared to the relative abundance of the diatom or is it high compared to other available prey items that were found from barcoding?

The high rate of infection refers to the high proportion of infected Nephrops individuals in our samples, with over 50% of the sampled individuals found to be infected by the parasitic dinoflagellate Hematodinium. This rate is not in reference to the relative abundance of the dinoflagellate in the area or in comparison to other available prey items detected through barcoding. We have clarified this in the text.

Line 321-322 If there are reports of cannibalism, why assume it is host contamination? Because of the higher value (98%)? Please elaborate more on why it was appropriate to treat it as host contamination since there is evidence of cannibalism in the species.

In our initial statement, we may not have adequately explained our rationale for considering these sequences as host contamination. The primary reason for treating the high proportion of Nephrops-derived sequences (over 98% of the data) as host contamination was the inability to confidently distinguish between true cannibalistic events and host contamination in our genetic dataset. Given this uncertainty, we opted for a conservative approach by disregarding these sequences to avoid potential overestimation of cannibalistic behavior in our study.

We have updated our statement in the manuscript as follows:

"Consequently, we generated high sequencing depth with over 18 million sequencing reads but found that over 98% of the data was assigned to Nephrops. Although there are reports of cannibalism between conspecifics (Sardà & Valladares, 1990), we could not confidently distinguish between true cannibalistic events and host contamination in our genetic dataset. Therefore, we conservatively treated these sequences as host contamination and disregarded them in our analysis."

We hope this clarification addresses your concern and provides a better understanding of our decision to treat Nephrops-derived sequences as host contamination in our study.

Minor comments:

Line 59: Should faeces be feces?

It is the British-English spelling.

Line 58-62 Consider switching the order of these two sentences.

Corrected.

Line 106 Add commas around “especially by females”

We appreciate the suggestion to add commas around "especially by females." However, we believe that the current sentence structure sufficiently conveys the intended meaning without additional emphasis. The context makes it clear that the suspension feeding strategy is particularly used by female Nephrops during the breeding period. As such, we have chosen to maintain the original sentence structure.

Line 106- When is the long breeding season? Does it correspond to when the samples were collected – see if they have a discussion on sex related and timing related prey identification

We have included information about the Nephrops breeding season, which lasts from late spring to early autumn. However, it is important to note that our samples were collected in January, outside of the breeding season. Therefore, we cannot make a direct connection between sex-related or timing-related prey identification and the breeding season in this study. We hope this clarifies any concerns regarding the timing of sample collection and its relation to the breeding season.

Line 107 Did the Santana et al. 2020 paper find this on samples collected during the long breeding time? It might be nice to include what time of year their samples came from for even further context about when they suspension feed.

Santana et al. (2020) collected their samples during the spring, which coincides with the breeding season. However, their study found no differences between male and female Nephrops in terms of their feeding habits during this time. We can include this information in the manuscript for further context on the suspension feeding behavior of Nephrops during their breeding season, as you suggested

Line 118-120 This sentence seems to have two separate ideas and is a little bit hard to follow. Can you please revise, perhaps break it into two separate sentences.

We have revised it to make it easier to understand and separated it into two sentences. The revised statement reads:

"Nephrops are a generalist forager and highly commercial benthic crustacean. Utilizing a molecular approach, we can consider them as unique natural biodiversity samplers, offering valuable insights into their ecosystem."

We believe that this revision clearly conveys the two ideas and should improve the readability of the text.

Line 133 Can you provide manufacturer details beyond just the website? Location of company? Same for BIOO Scientific on line 152.

We understand the importance of providing appropriate manufacturer details. However, since the companies for BIOO Scientific have changed and to avoid any confusion, we decided to remove the website link and BIOO Scientific from the text. We believe that readers can use their initiative to look up the updated information about the products mentioned in the manuscript.

Line 182 Before using MOTU, please define what that acronym means.

We define a MOTU at line 166 in the original draft of the manuscript at first instance of the Materials and methods section.

Line 190 Is that the same version of R as previously mentioned on line 183?

Yes but our reference here is not the R core environment, it’s to the package geomnet.

Line 204 What are the positive and negative controls?

Corrected.

Line 211 Can remove “With a modest number of remaining reads” and start the sentence with “We considered data…”.

Corrected.

Line 220 Add ‘and’ between 49% and 46%

Corrected.

---

## [Decision Letter · Decision Letter 1]

4 Jul 2023

PONE-D-23-04632R1DNA metabarcoding reveals the dietary profiles of a benthic marine crustacean, Nephrops norvegicusPLOS ONE

Dear Dr. Shum,

Thank you for re-submitting your manuscript to PLOS ONE. Both of the prior reviewers have made a second evaluation and one of these reviewers has made a few additional suggestions that are primarily editorial and do not affect the scientific value of the contribution, but I agree will make the contribution more scientifically sound. Therefore, I'd ask you to consider their suggestions as a final step to improve the manuscript and make it acceptable for publication. Taking into account these suggestions, please submit a revised version of the manuscript that addresses these final points raised during the review process.

Please submit your revised manuscript by Aug 18 2023 11:59PM. I don't expect to send it back to the reviewers again, and anticipate that I can send on a positive recommendation to the editorial office of the journal in the near future. However, if you will need more time than this to complete your revisions, please reply to this message or contact the journal office at plosone@plos.org. Please include the following items when submitting your revised manuscript:A rebuttal letter that responds to each point raised by the academic editor and reviewer(s). You should upload this letter as a separate file labeled 'Response to Reviewers'.A marked-up copy of your manuscript that highlights changes made to the original version. You should upload this as a separate file labeled 'Revised Manuscript with Track Changes'.An unmarked version of your revised paper without tracked changes. You should upload this as a separate file labeled 'Manuscript'.If applicable, we recommend that you deposit your laboratory protocols in protocols.io to enhance the reproducibility of your results. Protocols.io assigns your protocol its own identifier (DOI) so that it can be cited independently in the future. For instructions see: https://journals.plos.org/plosone/s/submission-guidelines#loc-laboratory-protocols. Additionally, PLOS ONE offers an option for publishing peer-reviewed Lab Protocol articles, which describe protocols hosted on protocols.io. Read more information on sharing protocols at https://plos.org/protocols?utm_medium=editorial-email&utm_source=authorletters&utm_campaign=protocols.

We look forward to receiving your revised manuscript.

Kind regards,

Lee W Cooper, Ph.D.

Section Editor

PLOS ONE

Journal Requirements:

Reviewers' comments:

Reviewer's Responses to Questions

**Comments to the Author**

1. If the authors have adequately addressed your comments raised in a previous round of review and you feel that this manuscript is now acceptable for publication, you may indicate that here to bypass the “Comments to the Author” section, enter your conflict of interest statement in the “Confidential to Editor” section, and submit your "Accept" recommendation.

Reviewer #1: (No Response)

Reviewer #2: All comments have been addressed

2. Is the manuscript technically sound, and do the data support the conclusions?

Reviewer #1: Yes

Reviewer #2: Yes

3. Has the statistical analysis been performed appropriately and rigorously? 

Reviewer #1: Yes

Reviewer #2: Yes

4. Have the authors made all data underlying the findings in their manuscript fully available?

Reviewer #1: Yes

Reviewer #2: Yes

5. Is the manuscript presented in an intelligible fashion and written in standard English?

Reviewer #1: Yes

Reviewer #2: Yes

6. Review Comments to the Author

Reviewer #1: Thank you for the chance to review this resubmission. I have a few minor editorial comments that should be addressed:

- The in-text citations need to be adjusted. There are unnecessary parentheses and the number order is inconsistent. For example ((46), (44)) should be (44, 46). Please correct this throughout the manuscript.

- I recommend abbreviating the site names for the North Sea. East North Sea to ENS and West North Sea to WNS. The way it is written at times makes it seem like you could be referring to a "West Sea". For examples, "the West and Irish Sea". There also several instances throughout where a direction (West of Ireland) is capitalised and does not need to be.

- Species should be italicised in the references.

- When referencing a specific paper in the text such as Santana et al. 2020, leave out the year and just insert the citation number. "Santana et al. (23) showed that...." See line 102 but there are at least one or two other occurrences.

- Line 100: put a period after "i.e., suspension feeding". Then start a new sentence "Suspension feeding is thought to be used especially by females..."

- Line 424: Should be "Southern Ocean" not "Antarctica ocean"

- Figure S2 was removed entirely rather than brought into the manuscript as was noted in a response to a previous comment. Was this intentional? I can't access the previous version or see tracked changes but believe the figures are the same as the original submission.

- In order to address concerns raised by both reviewers, it may also be useful to add a statement in the conclusion that suggests further research is needed to explore the role of some of the "secondary" or unexpected prey items found in the Nephrops diet in greater detail.

Reviewer #2: All of my original concerns have been addressed and this paper should be considered for acceptance.

7. PLOS authors have the option to publish the peer review history of their article (what does this mean?). If published, this will include your full peer review and any attached files.

Reviewer #1: **Yes: **Chelsea W. Koch

Reviewer #2: No

---

## [Author Response · Author response to Decision Letter 1]

12 Jul 2023

Thank you for re-submitting your manuscript to PLOS ONE. Both of the prior reviewers have made a second evaluation, and one of these reviewers has made a few additional suggestions that are primarily editorial and do not affect the scientific value of the contribution, but I agree will make the contribution more scientifically sound. Therefore, I ask you to consider their suggestions as a final step to improve the manuscript and make it acceptable for publication. Taking into account these suggestions, please submit a revised version of the manuscript that addresses these final points raised during the review process.

We have now evaluated the reviewers' comments (as per below) and believe the points raised are now satisfactory. In light of the review process, we acknowledge the reviewers anonymously for their comments.

Comments to the Author

Reviewer #1: Thank you for the chance to review this resubmission. I have a few minor editorial comments that should be addressed:

• The in-text citations need to be adjusted. There are unnecessary parentheses, and the number order is inconsistent. For example, [(46), (44)] should be [44, 46]. Please correct this throughout the manuscript.

We have made corrections with citations enclosed in square brackets [].

• I recommend abbreviating the site names for the North Sea. East North Sea to ENS and West North Sea to WNS. The way it is written at times makes it seem like you could be referring to a "West Sea". For example, "the West and Irish Sea". There also several instances throughout where a direction (West of Ireland) is capitalized and does not need to be.

We have taken this suggestion into account and corrected the names to North Sea East (NS-East) and North Sea West (NS-West).

• Species should be italicized in the references.

Corrected.

• When referencing a specific paper in the text such as Santana et al. 2020, leave out the year and just insert the citation number. "Santana et al. [23] showed that...." See line 102, but there are at least one or two other occurrences.

Corrected.

• Line 100: put a period after "i.e., suspension feeding". Then start a new sentence "Suspension feeding is thought to be used especially by females..."

Corrected.

• Line 424: Should be "Southern Ocean" not "Antarctica ocean"

Corrected.

• Figure S2 was removed entirely rather than brought into the manuscript as was noted in a response to a previous comment. Was this intentional? I can't access the previous version or see tracked changes but believe the figures are the same as the original submission.

This figure was implemented into Figure 2. It is cited as Figure 2B.

• In order to address concerns raised by both reviewers, it may also be useful to add a statement in the conclusion that suggests further research is needed to explore the role of some of the "secondary" or unexpected prey items found in the Nephrops diet in greater detail.

We have added a statement into the final sentence.

Reviewer #2: All of my original concerns have been addressed, and this paper should be considered for acceptance.

We thank the reviewer for their thorough evaluation and positive feedback.

---

## [Editor Report · Decision Letter 2]

14 Jul 2023

DNA metabarcoding reveals the dietary profiles of a benthic marine crustacean, Nephrops norvegicus

PONE-D-23-04632R2

Dear Dr. Shum,

Thank you for making those final changes to the manuscript. I am pleased to inform you that your manuscript has been judged scientifically suitable for publication and will be formally accepted for publication once it meets all outstanding technical requirements.

Kind regards,

Lee W Cooper, Ph.D.

Section Editor

PLOS ONE

---

## [Editor Report · Acceptance letter]

18 Jul 2023

PONE-D-23-04632R2 

DNA metabarcoding reveals the dietary profiles of a benthic marine crustacean, *Nephrops norvegicus*

Dear Dr. Shum:

I'm pleased to inform you that your manuscript has been deemed suitable for publication in PLOS ONE. Congratulations! Your manuscript is now with our production department. 

Kind regards, 

on behalf of

Dr. Lee W Cooper 

Section Editor

PLOS ONE